# The effect of lockdown on the outcomes of COVID-19 in Spain: An ecological study

**Camila Alves dos Santos Siqueira**[1], **Yan Nogueira Leite de Freitas**[2], **Marianna de Camargo Cancela**[3], **Monica Carvalho**[4], **Albert Oliveras-Fabregas**[5], **Dyego Leandro Bezerra de Souza**[1,5¤]*

**1** Graduate Program in Collective Health, Department of Collective Health, Federal University of Rio Grande do Norte, Natal, Brazil, **2** Federal University of Amazonas, Manaus, Brazil, **3** Division of Surveillance and Situation Analysis, Brazilian National Cancer Institute, Rio de Janeiro, Brazil, **4** Department of Renewable Energy Engineering, Federal University of Paraíba, João Pessoa, Brazil, **5** Research Group on Methodology, Methods, Models, and Outcomes of Health and Social Sciences (M₃O), Faculty of Health Sciences and Welfare, Center for Health and Social Care Research (CESS), University of Vic-Central University of Catalonia (UVic-UCC), Barcelona, Spain

¤ Current address: Carrer de la Sagrada Familia, Barcelona, Spain
* dysouz@yahoo.com.br

**Data Availability Statement:** This is an ecological-type observational study that utilized public domain data from the Health Ministry of Spain website. All

## Abstract

It is paramount to expand the knowledge base and minimize the consequences of the pandemic caused by the new Coronavirus (SARS-Cov2). Spain is among the most affected countries that declared a countrywide lockdown. An ecological study is presented herein, assessing the trends for incidence, mortality, hospitalizations, Intensive Care Unit admissions, and recoveries per autonomous community in Spain. Trends were evaluated by the Joinpoint software. The timeframe employed was when the lockdown was declared on March 14, 2020. Daily percentage changes were also calculated, with CI = 95% and p<0.05. An increase was detected, followed by reduction, for the evaluated indicators in most of the communities. Approximately 18.33 days were required for the mortality rates to decrease. The highest mortality rate was verified in Madrid (118.89 per 100,000 inhabitants) and the lowest in Melilla (2.31). The highest daily percentage increase in mortality occurred in Catalonia. Decreasing trends were identified after approximately two weeks of the institution of the lockdown by the government. Immediately the lockdown was declared, an increase of up to 33.96% deaths per day was verified in Catalonia. In contrast, Ceuta and Melilla presented significantly lower rates because they were still at the early stages of the pandemic at the moment of lockdown. The findings presented herein emphasize the importance of early and assertive decision-making to contain the pandemic.

## Introduction

The COVID-19 pandemic, a disease caused by a new coronavirus (SARS-CoV-2), has already directly affected more than 11,327,790 people around the world, causing 532,340 deaths until July 6 2020[1]. This scenario is even more concerning due to the inability of mass testing verified in many countries, indicating that the number of cases is potentially higher than the

data are publicity available. All details that might disclose the identity of the subjects were omitted or anonymized. In addition, all relevant data are within the manuscript and its Supporting Information files.

**Funding:** This work was financed by the "Coordenação de Aperfeiçoamento de Pessoal de Nível Superior" - Brazil (CAPES, Coordination for the Improvement of Higher Education Personnel) - financing code 001. The funding is intended to pay article-processing charges.

**Competing interests:** The authors have declared that no competing interests exist.

number of confirmed cases[2]. The experience of countries such as China, Italy, the United States, and Spain demonstrate that the Coronavirus Disease 2019 (COVID-19) has burdened health systems regardless of available investments and resources[3–6].

Without a vaccine or available pharmaceutical treatment, the actions to contain the dissemination of SARS-CoV-2 have been initially concentrated on isolation measures for confirmed cases and self-quarantine of those knowingly exposed. Differently from what was observed in 2012 with the Middle East respiratory syndrome (MERS) and in 2002 with the Severe Acute Respiratory Syndrome (SARS), isolation and quarantine were not sufficient to contain the dissemination of the new coronavirus. SARS-CoV-2 presents high transmissibility, from the onset of symptoms but also from asymptomatic cases of COVID-19[7,8].

Authorities and governments have adopted several forms of physical distancing as public health measures to contain the dissemination of the new coronavirus. Mass testing of the population, when possible, has also demonstrated to decrease the propagation of the virus efficiently. Physical distancing aims to avoid the social interaction of people, limiting mass gatherings by closing schools, public spaces, commercial establishments, and even non-essential workplaces. The objective of this strategy is to reduce the intensity peak of the epidemic curve ("flatten the curve"), decreasing the risk of health system collapse while simultaneously increasing the opportunity of developing studies focused on effective treatments and vaccines[7,9–11]. A more severe form of physical distancing is the lockdown, which is announced by authorities to restrict free movement in view of enforcing physical distancing and breaking the chains of transmission. Lockdowns prevent all public movement except essential services.

The experiences of Singapore, South Korea, and the territory of Hong Kong have demonstrated that physical distancing (although implemented in different degrees) and mass testing measures are effective in controlling the pandemic–especially if adopted correctly and in time [3,7,9]. Hong Kong endorsed serious physical distancing measurements and presented a COVID-19 mortality rate of approximately 0.38%, with only four deaths recorded due to the disease, well below global averages[1,12].

However, when facing a new pathogen, such as SARS-CoV-2, the public health measures adopted to date still generate debate among specialists[7,10,13]. Several mathematical models were developed to predict the impact of these measures on the course of the pandemic, health systems, and the economy of different countries[10,14–17].

In European countries, the recommendations regarding physical distancing vary, depending on how the pandemic advanced in each region. Italy was the first country to adopt physical distancing measures, and, in some areas, the way these measures were implemented generated a series of discussions between the authorities and the population[4,16]. In Spain, a lockdown was enforced on March 15, 2020, when the country presented 5753 confirmed cases and 136 deaths due to COVID-19[18]. On April 25, 2020, Spain started to ease the lockdown with a gradual lifting of restrictions due to decreasing trends in confirmed cases, hospitalizations, and daily deaths from the new coronavirus[19].

Considering that the first wave of the pandemic is losing strength in some European countries, mainly in Spain, it is necessary to evaluate the real impact of the physical distancing measures adopted hitherto. This valuable information can help authorities adopt evidence-based measures, potentially increasing the adherence of the population. Therefore, the objective of this study is to investigate the impact of physical distancing measures enforced by the autonomous communities of Spain, regarding incident cases, hospitalizations (general hospital ward and Intensive Care Unit ward), and mortality trends related to COVID-19.

## Materials and methods

An ecological study was carried out with aggregated data of the COVID-19 pandemic, available from the Health Ministry of Spain[20], for the period between March 14 and April 25, 2020. These dates corresponded to when the State of Alarm was declared by the government (which instituted the lockdown on March 15), and the day before the lockdown was eased (i.e., children allowed out if accompanied by a single adult), respectively[21].

Daily data was published by the Health Ministry of Spain, per autonomous community, on the number of new cases, hospital admissions, Intensive Care Unit (ICU) admissions, deaths, and recoveries[20]. Hospital admissions refer to the individuals admitted to general hospital wards. ICU admissions apply to the patients that received critical care in ICUs. Data for the development of this study were collected on April 26, 2020. Autonomous community population data were obtained from the Spanish National Statistics Institute for 2019[22]. All data are publicly available.

Information on new cases, hospital admissions, ICU admissions, deaths, and recoveries were analyzed by the *Joinpoint Regression Program*, version 4.8.0.1. This program assesses the trends throughout time, according to significant modifications in their evolution patterns.

Analysis of the evaluated indicators required the calculation of crude rates for each autonomous community and the autonomous cities of Ceuta and Melilla. Crude rates were calculated by dividing the number of daily observations by the population exposed to risk. Rates were expressed in terms of 100,000 inhabitants/day[23]. Calculations were carried out directly in the *Joinpoint* software. Stratified data analysis was carried out for each autonomous community. However, it was detected that some data did not follow the same pattern of accumulated data of the other communities. This was the case of Madrid and Castile-La Mancha, which precluded trend analysis for hospital and ICU admissions, and of Castile and León and Galicia, regarding ICU admissions.

Regarding trend analysis, the program identifies the joinpoint (time points in which the trend significantly changes) and calculates the percentage of change per time interval. The temporal unit employed herein was a day. For each indicator, the accumulated total crude rate was calculated for the period, and the daily crude rates were used to calculate the trends. For each segment, the *Daily Percentage Change (DPC)* was calculated to identify the statistical significance (p-value<0.05), with a 95% confidence level. The analysis was carried out considering an assumption of heteroscedasticity and variance of Poisson. Significant modifications in the curve represent the joinpoints. The connection of linear elements, by a graph, enables a succinct characterization of trends[24]. For the periods with a statistical significance of DPC, the trends can be classified as "increasing" or "decreasing". For those values with no statistical significance, the term "stable" was employed. Models with zero to three joinpoints were observed, and the model that presented the best fit with observed data was selected.

The analysis considered 43 days of monitoring, and incidence rates were calculated for the entire period for new confirmed cases, hospital admissions, ICU admissions, deaths, and recoveries. Calculations also included the mean, standard deviation, and median of the number of days elapsed since the state of alarm was declared by the government, along with the joinpoint that identified the change (from "increasing" to "decreasing") in the trend of the indicator.

## Results

During the period encompassed by this study (March 14 –April 25, 2020), 223,791 new cases of COVID-19 were registered in Spain, along with 23,135 deaths. Analysis of mortality trends, confirmed cases, hospital admissions, and ICU admissions revealed an increasing pattern,

followed by a reduction, for all regions with registries for these data. Ceuta was an exception, where stability was observed for the rates associated with hospital and ICU admissions. In Melilla, stable rates were identified for hospital admissions, and decreasing rates were obtained for ICU admissions. Figs 1, 2, 3 and 4 present daily COVID-19 rates for incidence, hospital admissions, ICU admissions, and mortality, respectively, per 100,000 inhabitants.

Ceuta and Melilla presented the lowest overall rates. Increasing recovery rates were obtained for most communities. Fig 5 shows that Asturias, Castile and León, Galicia, Balearic Islands Madrid, and Rioja presented increasing trends followed by a reduction. Among the indicators studied, mortality rates offered the best information quality, with little fluctuation throughout the data series.

The day when the trend changed varied across autonomous communities and evaluated indicators. Regarding the mean number of days elapsed for a change to occur in the evolution pattern of the disease and the beginning of the inflection of the curve, the mortality rate presented the latest joinpoint (mean = 18.33 days, standard deviation—SD = 5.37, median = 18.50 and interquartile range–IQR = 6). The following parameters were general hospital ward admissions (mean = 14.27 days, SD = 4.68, median = 14.00 and IQR = 2.5), ICU admissions (mean = 13.44 days, SD = 4.42, median = 13.00 and IQR = 5), and finally, incidences (mean = 12.18 days, SD = 2.92, median = 12.00 and IQR = 1.5). Change in trends, with a consequent reduction in the number of deaths, took longer than the national average for Navarre (34 days), Basque Country (25 days), Extremadura (24 days), Murcia (22 days), La Rioja (24 days), Ceuta (23 days), Andalusia (21 days), and Valencia and Balearic Islands (19 days each).

Tables 1 and 2 show that the communities with the highest rates of confirmed cases and mortality for the study period were: La Rioja, Madrid, and Castile-La Mancha. The lowest incidence trends occurred in the Canary Islands and Andalusia. Mortality rates were the highest in Madrid and La Rioja, while the lowest mortality rates were detected in Melilla and Ceuta. Regarding hospital and ICU admissions, the highest rates also occurred in the regions with a

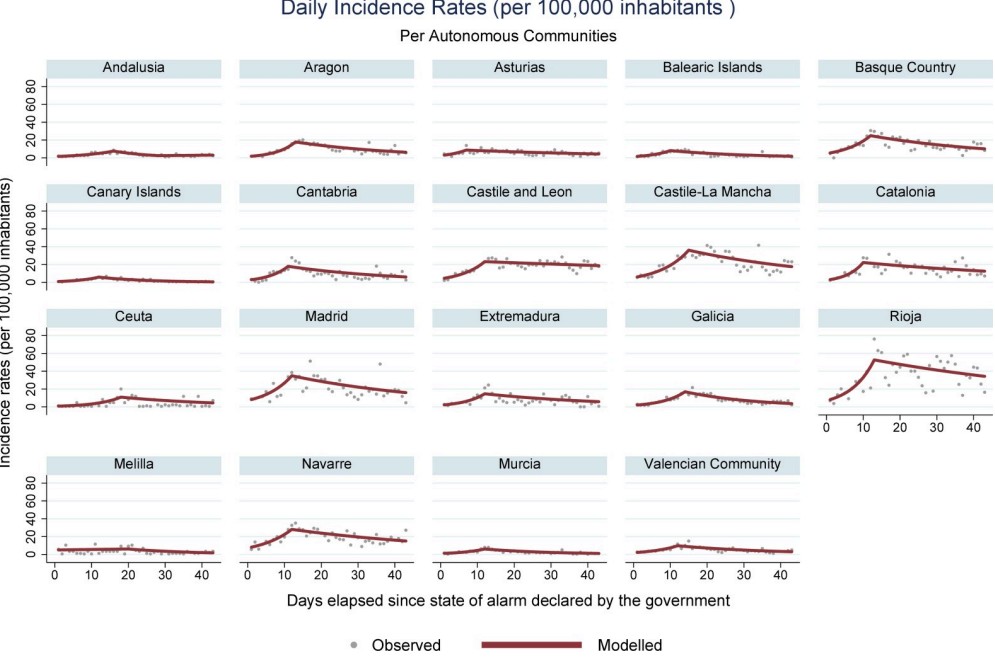

**Fig 1. COVID-19 daily incidence rates per 100,000 inhabitants in Spanish autonomous communities.**

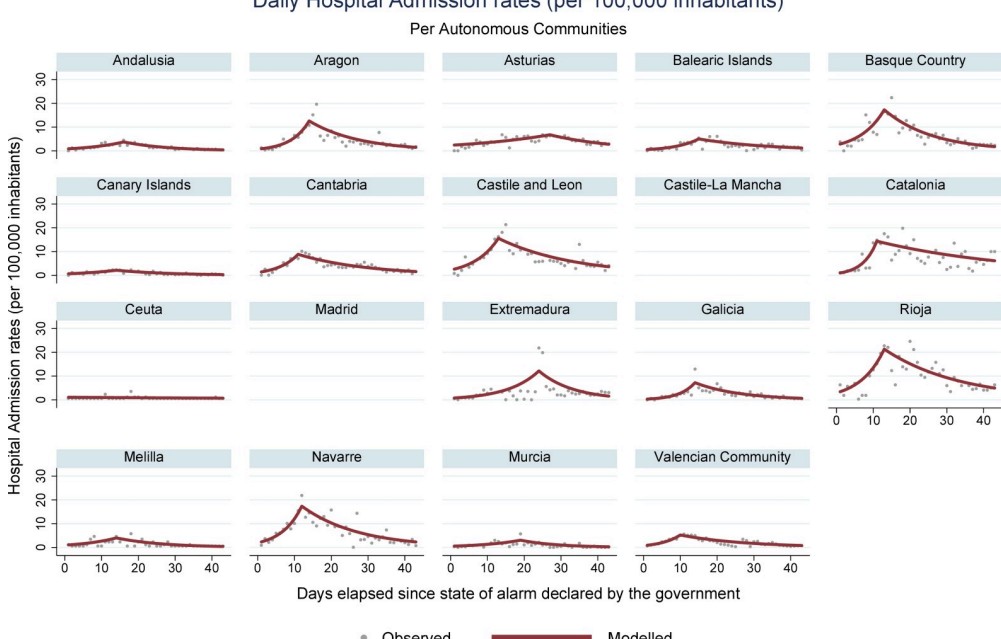

**Fig 2. COVID-19 daily hospital admission rates per 100,000 inhabitants in Spanish autonomous communities.**

higher number of cases: La Rioja is highlighted as the community with the highest hospital admission rates, while Catalonia presented the highest ICU admission rates. The recovery rates were also higher in the most affected locations, remarkably for La Rioja and Madrid.

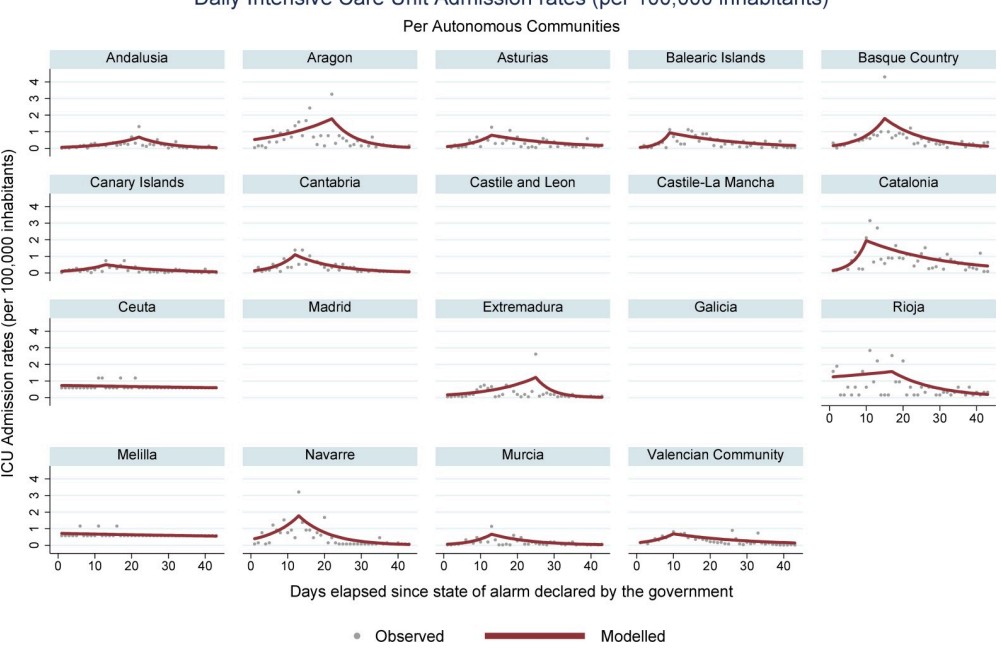

**Fig 3. COVID-19 daily Intensive Care Unit admission rates per 100,000 inhabitants in Spanish autonomous communities.**

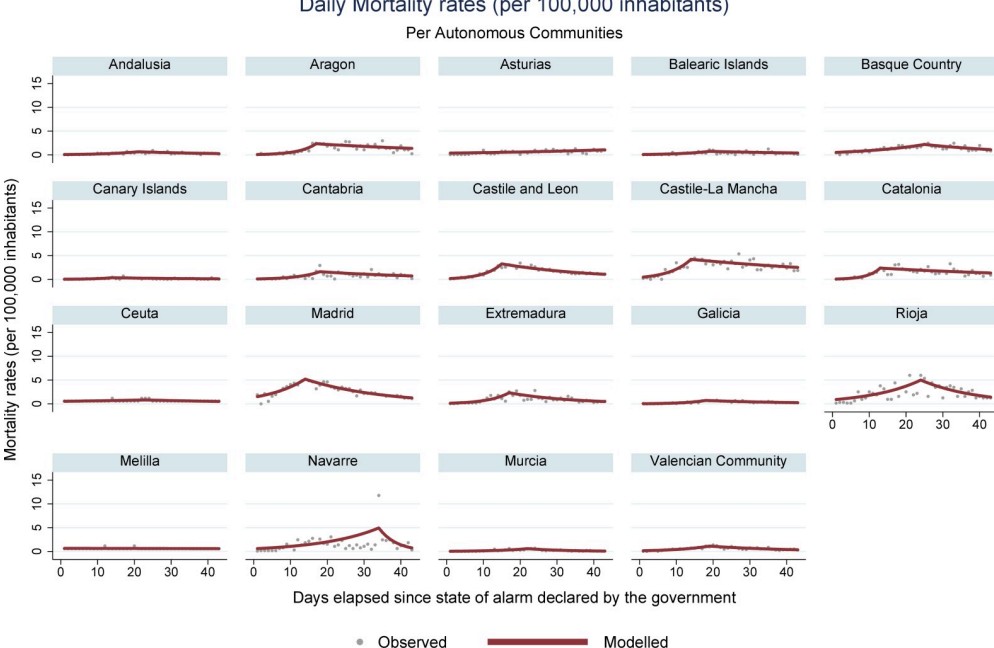

**Fig 4. COVID-19 daily mortality rates per 100,000 inhabitants in Spanish autonomous communities.**

Tables 1 and 2 also depict the DPC for each indicator and autonomous community. Catalonia is the community with the highest speed of increase in deaths per day. At the same time, Ceuta and Melilla presented the best situation in the context of the evolution of the disease.

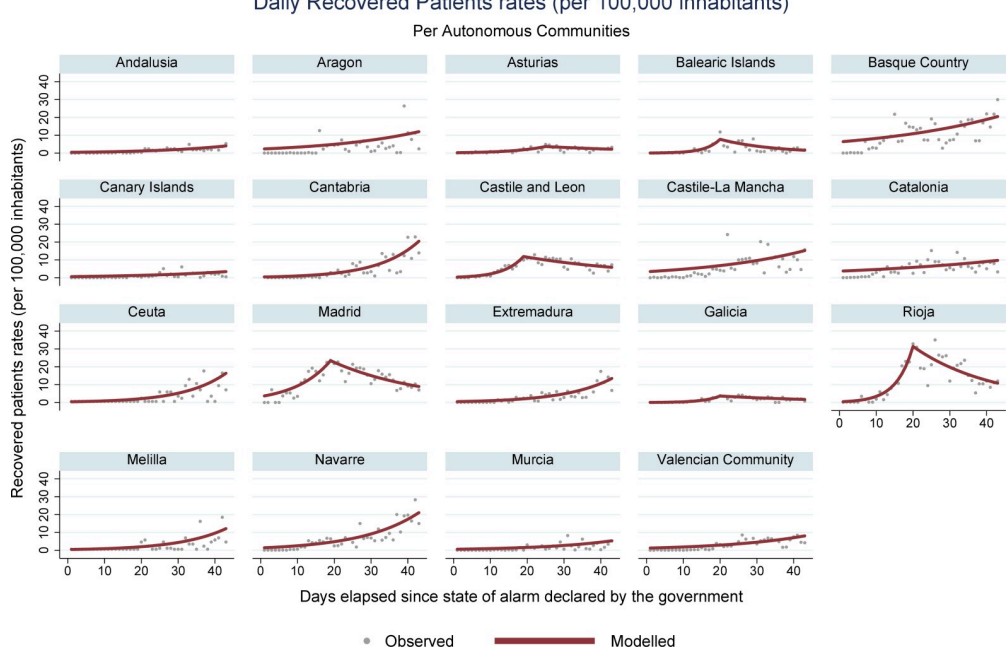

**Fig 5. COVID-19 daily recovered rates per 100,000 inhabitants in Spanish autonomous communities.**

**Table 1. Trends for COVID-19 incidence, hospital admission and ICU admission rates per Spanish autonomous community, 2020.**

| Autonomous Communities | | Incidence | | | | | | | Hospitalizations | | | | | ICU | | | | |
|---|---|---|---|---|---|---|---|---|---|---|---|---|---|---|---|---|---|---|
| Incidence | Population | n | Rates | DPC1 | JP | DPC2 | JP | DPC3 | n | Rates | DPC1 | JP | DPC2 | n | Rates | DPC1 | JP | DPC2 |
| Andalusia | 8,414,240 | 12,754 | 151.58 | 10.87* | 16 | -8.85* | 28 | 1.37 | 5656 | 67.22 | 10.10* | 16 | -8.00* | 709 | 8.43 | 11.82* | 22 | -11.23* |
| Aragon | 1,319,291 | 5335 | 404.38 | 21.04* | 13 | -3.42* | - | - | 2353 | 178.35 | 22.72* | 14 | -7.12* | 304 | 23.04 | 5.92* | 22 | -14.93* |
| Asturias | 1,022,800 | 2509 | 245.31 | 19.35 | 7 | -1.95* | - | - | 1774 | 173.45 | 4.01* | 27 | -5.42* | 132 | 12.91 | 17.90* | 13 | -5.06* |
| Balearic Islands | 1,149,460 | 1889 | 164.34 | 19.25* | 10 | -4.51* | - | - | 1053 | 91.61 | 16.98* | 15 | -4.96* | 165 | 14.35 | 42.53* | 9 | -4.96* |
| Basque Country | 2,207,776 | 13,794 | 624.79 | 15.65* | 12 | -2.87* | - | - | 13,794 | 624.79 | 15.65* | 12 | -2.87* | 1216 | 55.08 | 6.16* | 25 | -3.74* |
| Canary Islands | 2,153,389 | 2077 | 96.45 | 17.90* | 12 | -7.28* | - | - | 865 | 40.17 | 9.16* | 14 | -6.46* | 165 | 7.66 | 14.24* | 13 | -6.37* |
| Cantabria | 581,078 | 2299 | 395.64 | 19.88* | 11 | -3.34* | - | - | 988 | 170.03 | 19.79* | 11 | -5.18* | - | - | - | - | - |
| Castile and Leon | 2,399,548 | 18,515 | 771.60 | 15.85* | 12 | -0.69 | - | - | 7628 | 317.89 | 16.08* | 13 | -4.75* | - | - | - | - | - |
| Castile La Mancha | 2,032,863 | 18,706 | 920.18 | 13.53* | 15 | -2.52* | - | - | - | - | - | - | - | - | - | - | - | - |
| Catalonia | 7,675,217 | 47,280 | 616.01 | 24.62* | 10 | -1.72* | - | - | 25,465 | 331.78 | 30.36* | 11 | -2.62* | 2554 | 33.28 | 33.32* | 10 | -4.56* |
| Ceuta | 84,777 | 131 | 154.52 | 16.42* | 18 | -3.51 | - | - | 10 | 11.80 | -0.98 | - | - | 4 | 4.72 | -0.5 | - | - |
| Madrid (Community) | 6,663,394 | 62,817 | 942.72 | 14.14* | 12 | -2.48* | - | - | - | - | - | - | - | - | - | - | - | - |
| Extremadura | 1,067,710 | 3399 | 318.34 | 18.68* | 12 | -2.94* | - | - | 1513 | 141.71 | 12.55* | 24 | -10.20* | 114 | 10.68 | 8.63* | 25 | -20.71* |
| Galicia | 2,699,499 | 9176 | 339.91 | 17.58* | 14 | -5.15* | - | - | 2735 | 101.32 | 26.65* | 14 | -7.92* | 181 | 6.70 | 24.79* | 12 | -10.84* |
| La Rioja | 316,798 | 4720 | 1489.91 | 16.85* | 13 | -1.42* | - | - | 1342 | 423.61 | 16.48* | 13 | -4.75* | 82 | 25.88 | 1.43 | 17 | -7.80* |
| Melila | 86,487 | 118 | 136.44 | 0.97 | 20 | -5.25* | - | - | 41 | 47.41 | 10.80* | 14 | -7.62* | 3 | 3.47 | -0.61* | - | - |
| Navarre | 654,214 | 5306 | 811.05 | 11.96* | 12 | -1.99* | - | - | 1914 | 292.56 | 20.07* | 12 | -6.31* | 125 | 19.11 | 13.21* | 13 | -11.14 |
| Murcia | 1,493,898 | 1724 | 115.40 | 15.99* | 12 | -5.34* | - | - | 619 | 41.44 | 9.88* | 19 | -9.56* | 104 | 6.96 | 21.29* | 13 | -8.84* |
| Valencian (Community) | 5,003,769 | 11,242 | 224.67 | 14.30* | 12 | -3.72* | - | - | 4706 | 94.05 | 21.35* | 10 | -5.50* | 625 | 12.49 | 17.14* | 10 | -4.66* |

JP = Joinpoint (days); DPC = daily percentage change

*Statistically significant data.

## Discussion

The results of the COVID-19 data analysis in Spain demonstrate the positive impact of the lockdown in containing the disease. It was possible to identify a similar pattern in the majority of autonomous communities in Spain, characterized by a pronounced decline in incidence, hospital admissions, ICU admissions, and mortality rates. The best indicator for the evaluation of the consequences of the pandemic was the mortality rate, which presented the highest uniformity across registries, besides representing the worst outcome of the disease. The assessment of these trends is a vital instrument to substantiate decision-making[16].

Different countries in different continents have enforced non-pharmaceutical measures, of which the lockdown can be highlighted, to contain the propagation of the COVID-19 pandemic[13,16]. Lockdowns enable the health system to increase its assistance capacity and reduce the transmission of the disease by about 60%, considering it affects symptomatic and asymptomatic patients[11].

A study that evaluated pandemic data in Spain, between February 24 and April 5, 2020, identified that physical distancing measures were effective to control the spreading of the disease[25,26], especially when correctly enforced and with adequate duration[27]. Scientific literature also reports similar results to those presented herein, in countries that effectively controlled the pandemic by adopting these restrictive measures: Singapore, South Korea, and the territory of Hong Kong[3,7,9], as well as China[28,29].

This study analyzed data after the lockdown was enforced by the Spanish government, on March 14, 2020, until the beginning of the flexibilization of social restrictions. The state of

**Table 2. Trends for COVID-19 mortality and recovery rates per Spanish autonomous community, 2020.**

| Autonomous Communities | | Mortality | | | | | Recovered | | | | |
|---|---|---|---|---|---|---|---|---|---|---|---|
| Incidence | Population | n | Rates | DPC1 | JP | DPC2 | n | Rates | DPC1 | JP | DPC2 |
| Andalusia | 8,414,240 | 1143 | 13.58 | 12.01* | 21 | -4.06* | 4746 | 56.40 | 5.43* | - | - |
| Aragon | 1,319,291 | 705 | 53.44 | 23.80* | 17 | -2.06* | 1960 | 148.56 | 3.96* | - | - |
| Asturias | 1,022,800 | 248 | 24.25 | 2.51* | - | - | 748 | 73.13 | 12.20* | 25 | -2.90* |
| Balearic Islands | 1,149,460 | 174 | 15.14 | 13.19* | 19 | -2.69 | 1122 | 97.61 | 28.88* | 20 | -6.66* |
| Basque Country | 2,207,776 | 1216 | 55.08 | 6.16* | 25 | -3.74* | 9597 | 434.69 | 2.80* | - | - |
| Canary Islands | 2,153,389 | 130 | 6.04 | 21.70* | 14 | -3.87* | 1120 | 192.75 | 9.44* | - | - |
| Cantabria | 581,078 | 183 | 31.49 | 18.28* | 18 | -3.14* | 6206 | 258.63 | 20.81* | 19 | -2.91* |
| Castile and Leon | 2,399,548 | 1665 | 69.39 | 23.22* | 15 | -3.95* | 5191 | 255.35 | 3.55* | - | - |
| Castile La Mancha | 2,032,863 | 2324 | 114.32 | 18.39* | 14 | 1.76* | 1043 | 48.44 | 4.05* | - | - |
| Catalonia | 7,675,217 | 4553 | 59.32 | 33.96* | 13 | -1.88* | 16,974 | 221.15 | 10.50* | - | - |
| Ceuta | 84,777 | 4 | 4.72 | 1.90* | 23 | -2.35* | 105 | 123.85 | 9.23* | - | - |
| Madrid (Community) | 6,663,394 | 7922 | 118.89 | 9.91* | 14 | -4.77* | 35,377 | 530.92 | 10.83* | 19 | -3.93* |
| Extremadura | 1,067,710 | 420 | 39.34 | 19.30* | 17 | -5.80* | 1580 | 147.98 | 8.18* | - | - |
| Galicia | 2,699,499 | 394 | 14.6 | 19.00* | 18 | -4.06* | 3262 | 120.84 | 24.76* | 20 | -3.18* |
| La Rioja | 316,798 | 312 | 98.49 | 7.62* | 24 | -6.40* | 2036 | 642.68 | 24.83* | 20 | -4.52* |
| Melila | 86,487 | 2 | 2.31 | -0.24 | - | - | 81 | 93.66 | 8.18* | - | - |
| Navarre | 654,214 | 431 | 65.88 | 6.61* | 34 | -19.00* | 1835 | 280.49 | 6.64* | - | - |
| Murcia | 1,493,898 | 127 | 8.50 | 13.02* | 22 | -9.51* | 920 | 61.58 | 5.34* | - | - |
| Valencian (Community) | 5,003,769 | 1182 | 23.62 | 11.84* | 19 | -4.73* | 6241 | 124.73 | 4.54* | - | - |

JP = Joinpoint (days); DPC = daily percentage change

*Statistically significant data.

alarm declared by the government centralized the decisions of the country, even though the Spanish Healthcare System is organized per autonomous communities, which generated criticism from some autonomous communities[21].

The lockdown was implemented in Spain through inter-sector actions that involved sanitary institutions, armed forces, and non-governmental organizations, to reduce the mobility of people. In this context, an important instrument to evaluate this reduction is the "COVID-19 Community Mobility Report", created by Google[30], with data on the movement of people, comparing the mobility of people before and after the pandemic, considering the governmental measures instituted in the world.

The lockdown instituted in Spain resulted in a reduction of mobility, registered by Google [30]. For the period considered herein, it was possible to identify decreases in mobility: -92% in retail stores and recreation, -66% in grocer's shops and pharmacies, -77% in parks, -82% in public transportation stations, and -62% in workplaces. The only criterion that presented an increase in social mobility referred to residential areas: there was an increase of 21% due to the concentration of population in their households and neighborhoods. These mean values reflect the effect of national mobility restrictions, which could have varied across autonomous communities along with the local measures implemented before the institution of the national lockdown[6].

The highest accumulated incidence trends were detected in La Rioja and Madrid. La Rioja also presented high rates of hospital and ICU admissions, as well as the Basque Country and Catalonia. Madrid, the Basque Country, Navarre, and Catalonia are the autonomous communities that concentrate the most important urban centers and present the highest GDP per inhabitant in Spain, with more developed transportation systems[31]. These characteristics

probably have helped spread the SARS-CoV-2. The behavior of incidence trends curves was similar to the curves of hospital and ICU admissions, accompanying the same increasing patterns throughout the study period, with posterior decrease.

In Ceuta, there was stability for the rates of hospital admissions and ICU admissions, and for Melilla, mortality rates were stable, with decreasing rates for ICU admissions. These communities were the least affected, probably because the lockdown was enforced at an early stage of the pandemic in these locations. It must be highlighted that both cities are located in the North region of the African continent, where the first cases were identified later than in the Iberian Peninsula.

Regarding mortality, it was observed that the communities of Madrid (118.90 deaths per 100,000 inhabitants) and Castile-La Mancha (114.32) presented the highest accumulated rates in the study period. It is discussed, in light of these data, whether the protests in celebration of Women's Day (which occurred on March 8, 2020 mainly in Madrid) could have contributed to increasing the dissemination of SARS-CoV-2 during the evaluated period[32]. However, it is difficult to establish an association with this specific event because, at the beginning of March, several mass gathering events continued to be held around the country.

Regarding changes in trend patterns, mortality rates required more time to present a shift in its curves. This information is crucial because public policies directed to isolation and control of a pandemic with similar characteristics must consider this time gap until results can be identified[11].

The Spanish communities that required more time than the national average (18.33 days) to reduce the daily number of deaths were the Basque Country, Extremadura, Murcia, La Rioja, Ceuta, Navarre, Andalusia, Castile and León, and Valencia. The delay for a change to be observed in the curve of these regions can be associated with the transmission of the disease by asymptomatic patients in allowed environments, such as supermarkets and pharmacies. Also, the infection of health professionals and other essential categories must be mentioned, along with late deaths of acute patients hospitalized in ICUs in more affected areas. In Navarre, an outlier was identified in the number of deaths on April 16, 2020, which dislocated the joinpoint to the 34th day. This probably reflects *a posteriori* notification of deaths occurred outside the hospital environment.

There was a high number of deaths in nursing homes in Spain. This part of the population, generally with advanced age and multiple comorbidities, is the most vulnerable to COVID-19 [33]. It is estimated that the mortality rate is four times higher than for non-institutionalized individuals. The institutionalized population must be a priority for preventive actions due to their vulnerability to respiratory diseases and the coexistence of several people in small, common spaces[34]. Another population at risk includes health professionals. The number of infected health professionals must be the lowest possible to minimize adverse effects in the quality of health assistance, aimed at the effective control of the pandemic[6,35,36].

The rate of recoveries, which presented increases for most communities, was higher in more affected locations, especially in La Rioja and Madrid, possibly due to the recovery of patients that were already ill. For the communities of Asturias, Castile and León, Galicia, Balearic Islands, Madrid, and La Rioja, the initially increasing trends were followed by decreasing trends for recoveries, in a similar pattern to the incidence curve. Data published by the Spanish Ministry of Health does not specify how the recovered cases are accounted for. This indicator is reported in almost all COVID-19 statistics but must be standardized to enable comparison across different territories.

Another point that must be discussed, besides the time required for a change in trends, is the speed of increase of all aforementioned indicators, even after the lockdown was declared throughout Spain. This was observed by analysis of the DPC. Catalonia presented the highest

DPC, with a daily increase of 24.62% in cases, and 33.96% in deaths, before the trend pattern changed. This finding corroborates scientific literature results and highlights the importance of enforcing early physical distancing to reduce the future consequences in the number of cases and deaths[16,25,37], as observed in Germany[16]. In contrast, the community of Ceuta presented an increase of only 1.9% in deaths, and Melilla exhibited a stable pattern regarding new cases and deaths. These two communities experienced early lockdowns, enforced at the early stages of the pandemic, which visibly mitigated consequences. In the cases of these two regions, it would probably not be necessary to institute such a long lockdown time. Relaxation of measures could have started before the national territory, along with other isolated regions such as La Gomera in the Canary Islands and some rural zones of the peninsula. This could have helped reduce the economic impact on these areas.

Scientific literature also remarks that the Spanish reality could be experienced by other countries. Intermittent physical distancing measures are a possibility for the following months, according to the capacity of each healthcare system and development of an effective treatment or vaccine[9]. These physical distancing restrictions, however, accompany several drawbacks that cannot be neglected, especially concerning economic impacts. Low-income individuals face difficulties in staying at home and support themselves during the lockdown period[8,13]. Also, other vulnerable groups must be considered, such as the homeless, the imprisoned, the elderly, those with special needs, and illegal immigrants[13].

This study emphasizes the importance of making scientifically-based decisions and adopting broad public policies that do not favor any specific social group. Scientific evidence is crucial to ensure the population that the best decisions are being implemented to combat the pandemic[11,13]. Scientific evidence is also necessary to adequately address the economic crisis that developed as a consequence of the adopted measures[11]. One of the strengths of this study is the comparison of populations in different moments of the pandemic, with varying rates of incidence, but submitted to the same public health measures during the same length of time. Therefore, the findings herein presented can subsidize the decision-making process in different international contexts.

Regarding the limitations, the constant review of data must be mentioned in the pandemic context, with the possibilities of delayed notifications according to the information made available by each autonomous community. An eventual accumulation of data for subsequent publication would hinder the evaluation of daily trends, due to the probabilities of under-notification. In this way, it is essential to interpret data with caution mainly because of under-notification, as approximately one-third of the cases that occurred in Spain were asymptomatic, plus the high number of people that were sick but not tested[38]. Also, it is essential to remark that the majority of diagnoses in Spain followed PCR tests. Therefore, despite its limitations, the epidemiological importance of this study is undeniable.

In conclusion, it was possible to identify a change in COVID-19 data trends in Spain, with a reduction in rates after two or three weeks of the institution of the lockdown by the government[21]. Even with the enforcement of the lockdown, some communities experienced increases in daily mortality rates over 20%, such as Catalonia and Aragon, reaching a maximum of 34% increase in the daily number of deaths in Catalonia. The higher rates in these communities are possibly associated with the higher demographic density of the most affected cities and higher mobility before the lockdown. The communities with the lowest quantity of cases, among which Ceuta and Melilla, in contrast, practically did not suffer increases in mortality, emphasizing the benefits of early lockdowns.

Finally, the current scientific literature is assertive regarding the importance of physical distancing enforcements, especially in the absence of effective treatments or vaccines. However, more studies are still required, focusing on the real impact of physical distancing, especially

considering health indicators. A better comprehension of the importance of policies that enforce lockdowns is vital to contain the pandemic and reduce the future consequences of its evolution in different countries.

## Author Contributions

**Conceptualization:** Marianna de Camargo Cancela, Dyego Leandro Bezerra de Souza.

**Data curation:** Marianna de Camargo Cancela, Albert Oliveras-Fabregas, Dyego Leandro Bezerra de Souza.

**Methodology:** Marianna de Camargo Cancela, Dyego Leandro Bezerra de Souza.

**Writing – original draft:** Camila Alves dos Santos Siqueira, Yan Nogueira Leite de Freitas, Monica Carvalho.

**Writing – review & editing:** Camila Alves dos Santos Siqueira, Yan Nogueira Leite de Freitas, Marianna de Camargo Cancela, Monica Carvalho, Albert Oliveras-Fabregas, Dyego Leandro Bezerra de Souza.

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
