## [Decision Letter · Decision Letter 0]

2 Jul 2020

PONE-D-20-16051

The effect of lockdown on the outcomes of COVID-19 in Spain: an ecological study

PLOS ONE

Dear Dr. Dyego Leandro Bezerra de Souza

Thank you for submitting your manuscript to PLOS ONE. After careful consideration, we feel that it has merit but does not fully meet PLOS ONE’s publication criteria as it currently stands. Therefore, we invite you to submit a revised version of the manuscript that addresses the points raised during the review process.

I read your article with interest.

I believe that, following the reviewer's suggestion, the document can be accepted

We look forward to receiving your revised manuscript.

Kind regards,

Francesco Di Gennaro

Academic Editor

PLOS ONE

Additional Editor Comments:

I read your article with interest.

I believe that, following the reviewer's suggestion, the document can be accepted

Reviewers' comments:

Reviewer's Responses to Questions

**Comments to the Author**

1. Is the manuscript technically sound, and do the data support the conclusions?

Reviewer #1: Yes

Reviewer #2: Partly

2. Has the statistical analysis been performed appropriately and rigorously? 

Reviewer #1: Yes

Reviewer #2: N/A

3. Have the authors made all data underlying the findings in their manuscript fully available?

Reviewer #1: Yes

Reviewer #2: Yes

4. Is the manuscript presented in an intelligible fashion and written in standard English?

Reviewer #1: Yes

Reviewer #2: Yes

5. Review Comments to the Author

Reviewer #1: Summary of research:

Overall, this is a very important and well conceptualized, timely study whose findings will be useful for guiding public health policy globally.

Major:

The major observation is that the discussion did not compare findings with work from previous authors. The authors mention that South Korea, China and Singapore instituted lock downs but do not share how the findings of this study compare with what happened in the other countries. The authors should go back to literature and explain how their findings compare with the trends in other countries that had the outbreak before Spain.

Minor:

There are minor grammatical errors, possibly due to the article being translated from Spanish but these do not affect the general understand-ability of the paper.

In the abstract, delete the comma between 'countries' and 'which declared'

Line 33, delete the comma between 'countries' and 'which declared'

Line 45: Use the word 'in contrast' instead of 'in opposition'

Line 47: Rewrite the sentence as: "This highlights the importance of early and assertive..." and make similar changes in the conclusion in the abstract.

Line 60: The word independently doesn't bring out the meaning, perhaps use another word.

Line 67: Remove all commas

The figures were well illustrated and are easy to understand. The methods are sound and the statistical methods employed are appropriate. The results and data are well presented and can be relied on because they are from an official source (Ministry of Health website). The references are okay but there is need to compare with other countries, this study has mainly focused on Europe. It would be good to see if the same effect was observed in Asia.

The research title is precise and concise and describes exactly what the authors did.

Reviewer #2: The text was written correctly. The analysis of essential elements to clarify the factors influencing mortality in the various areas of Spain is lacking (discussion about the use of different IPD based on the geographical area? Mobility of the area? Number of health workers infected? Diagnostic methods of investigation?). English was good.

6. PLOS authors have the option to publish the peer review history of their article (what does this mean?). If published, this will include your full peer review and any attached files.

Reviewer #1: **Yes: **Joanitah Atuhaire-Mutanga

Reviewer #2: No

---

## [Author Response · Author response to Decision Letter 0]

6 Jul 2020

PlosOne

Manuscript PONE-D-20-16051

The effect of lockdown on the outcomes of COVID-19 in Spain: an ecological study

Dear Editor,

We would like to take this opportunity to thank all the anonymous reviewers that have taken part in this review process for their valuable and detailed comments. We feel that their input has considerably improved the manuscript. All comments were relevant and the new, revised version of the manuscript followed the PlosOne recommendations.

The authors have provided the detailed answers to each Reviewers’ comments in the following. A copy of the manuscript with tracked changes is also provided, so the reviewers can verify each change implemented.

The authors have updated the numbers regarding the number of cases and deaths, as of July 5 2020. And we have also employed the term “physical distancing” as recommended by the World Health Organization (instead of “social distancing”), although the meaning is the same.

Reviewers' comments:

Reviewer #1: Summary of research:

Overall, this is a very important and well conceptualized, timely study whose findings will be useful for guiding public health policy globally.

Major:

The major observation is that the discussion did not compare findings with work from previous authors. The authors mention that South Korea, China and Singapore instituted lock downs but do not share how the findings of this study compare with what happened in the other countries. The authors should go back to literature and explain how their findings compare with the trends in other countries that had the outbreak before Spain.

Authors’ answer: We have revised the scientific literature and added comparisons that explain and support our findings, from the trends studied in other countries.

Minor:

There are minor grammatical errors, possibly due to the article being translated from Spanish but these do not affect the general understand-ability of the paper.

Authors’ answer: The new, revised version of the manuscript has been thoroughly reviewed by a professional English proof-reader.

In the abstract, delete the comma between 'countries' and 'which declared'

Line 33, delete the comma between 'countries' and 'which declared'

Authors’ answer: The sentence has been changed. 

Spain is among the most affected countries that declared a country-wide lockdown.

Line 45: Use the word 'in contrast' instead of 'in opposition'

Authors’ answer: The term has been substituted, as indicated. 

In contrast, Ceuta and Melilla presented significantly lower rates because they were still at the early stages of the pandemic at the moment of lockdown.

Line 47: Rewrite the sentence as: "This highlights the importance of early and assertive..." and make similar changes in the conclusion in the abstract.

Authors’ answer: The sentence has been rewritten, following your suggestion:

The findings presented herein emphasize the importance of early and assertive decision-making to contain the pandemic.

Line 60: The word independently doesn't bring out the meaning, perhaps use another word.

Authors’ answer: The sentence has been rewritten, 

…has burdened health systems regardless of available investments and resources.

Line 67: Remove all commas

Authors’ answer: All commas were removed and the sentence was slightly changed. 

…isolation and quarantine were not sufficient to contain the dissemination of the new coronavirus.

The figures were well illustrated and are easy to understand. The methods are sound and the statistical methods employed are appropriate. The results and data are well presented and can be relied on because they are from an official source (Ministry of Health website). The references are okay but there is need to compare with other countries, this study has mainly focused on Europe. It would be good to see if the same effect was observed in Asia.

Authors’ answers: We have revised the scientific literature and added comparisons that explain and support our findings, from the trends studied in other countries.

The authors wish to thank you for the time dedicated to reviewing our manuscript. We believe the new, revised version, is much better. The manuscript has undergone a throughout, detailed review to improve clarity and understandability.

Reviewer #2: The text was written correctly. The analysis of essential elements to clarify the factors influencing mortality in the various areas of Spain is lacking (discussion about the use of different IPD based on the geographical area? Mobility of the area? Number of health workers infected? Diagnostic methods of investigation?). English was good.

Authors’ answers: Thanks for your comments. The reviewer is right, addressing the points mentioned is crucial for the correct comprehension of the results presented. We have added text that approaches these aspects in the discussion section. 

The authors wish to thank you for the time dedicated to reviewing our manuscript. We believe the new, revised version, is much better. The manuscript has undergone a throughout, detailed review to improve clarity and understandability.

Sincerely,

Dyego Leandro Bezerra de Souza dysouz@yahoo.com.br

Current Address: Carrer de la Sagrada Familia, 7, 08500 Vic, Barcelona, Spain. Telephone: +34 640169398 Research Group on Methodology, Methods, Models and Outcomes of Health and Social Sciences (M3O). Faculty of Health Sciences and Welfare. Centre for Health and Social Care Research (CESS). University of Vic-Central University of Catalonia (UVic-UCC)

---

## [Decision Letter · Decision Letter 1]

15 Jul 2020

The effect of lockdown on the outcomes of COVID-19 in Spain: an ecological study

PONE-D-20-16051R1

Dear Dr. Dyego Leandro Bezerra de Souza ,

We’re pleased to inform you that your manuscript has been judged scientifically suitable for publication and will be formally accepted for publication once it meets all outstanding technical requirements.

Kind regards,

Francesco Di Gennaro

Academic Editor

PLOS ONE

Additional Editor Comments (optional):

Congratulations, well done.

Reviewers' comments:

Reviewer's Responses to Questions

**Comments to the Author**

1. If the authors have adequately addressed your comments raised in a previous round of review and you feel that this manuscript is now acceptable for publication, you may indicate that here to bypass the “Comments to the Author” section, enter your conflict of interest statement in the “Confidential to Editor” section, and submit your "Accept" recommendation.

Reviewer #1: All comments have been addressed

Reviewer #2: All comments have been addressed

2. Is the manuscript technically sound, and do the data support the conclusions?

Reviewer #1: Yes

Reviewer #2: Partly

3. Has the statistical analysis been performed appropriately and rigorously? 

Reviewer #1: Yes

Reviewer #2: Yes

4. Have the authors made all data underlying the findings in their manuscript fully available?

Reviewer #1: Yes

Reviewer #2: Yes

5. Is the manuscript presented in an intelligible fashion and written in standard English?

Reviewer #1: Yes

Reviewer #2: Yes

6. Review Comments to the Author

Reviewer #1: The authors have addressed the comments and the article has improved significantly, I wish them the best.

Reviewer #2: (No Response)

7. PLOS authors have the option to publish the peer review history of their article (what does this mean?). If published, this will include your full peer review and any attached files.

Reviewer #1: **Yes: **Joanitah Atuhaire-Mutanga

Reviewer #2: No

---

## [Editor Report · Acceptance letter]

20 Jul 2020

PONE-D-20-16051R1 

The effect of lockdown on the outcomes of COVID-19 in Spain: an ecological study 

Dear Dr. de Souza:

I'm pleased to inform you that your manuscript has been deemed suitable for publication in PLOS ONE. Congratulations! Your manuscript is now with our production department. 

Kind regards, 

on behalf of

Dr. Francesco Di Gennaro 

Academic Editor

PLOS ONE